# First Case Report of Uterine Leiomyosarcoma Diagnosed After Transcervical Fibroid Ablation

**DOI:** 10.3390/jcm14010088

**Published:** 2024-12-27

**Authors:** Dimitrios Chronas, Inna Jörg, Kristina Bolten, Laura Reich, David Toub, Zsuzsanna Varga

**Affiliations:** 1Spital Zollikerberg, 8125 Zollikerberg, Switzerland; dimitrios.chronas@spitalzollikerberg.ch (D.C.);; 2Gynesonics®, Redwood City, CA 94063, USA; dtoub@gynesonics.com; 3Institut für Pathologie und Molekularpathologie, Universitätsspital Zürich, 8091 Zürich, Switzerland; zsuzsanna.varga@usz.ch

**Keywords:** uterine fibroids, STUMP, leiomyosarcoma, transcervical radiofrequency ablation

## Abstract

**Background:** Uterine fibroids are benign monoclonal neoplasms of the myometrium, representing the most common female pelvic neoplasms globally. Treatments may be invasive, such as hysterectomy and myomectomy, non-invasive, such as medical therapy or focused ultrasound, or minimally invasive, such as transcervical radiofrequency ablation (TFA). To date, more than 12,000 women have been treated worldwide using TFA with the Sonata^®^ System. **Case Presentation:** We present the first case report of TFA on a presumptive fibroid that was initially reclassified as a STUMP (smooth muscle tumor of uncertain malignant potential) and, after additional surgical treatment, leiomyosarcoma. **Conclusion:** This case highlights that, while uterine sarcoma is rare, inadvertent treatment may still result due to a lack of reliable diagnostic modalities. Nonetheless, TFA with the Sonata System represents a minimally invasive option that might not alter the prognosis of an undiagnosed uterine sarcoma as this treatment is not intraperitoneal and does not resect/morcellate tissue.

## 1. Introduction

Uterine fibroids (leiomyomata) are benign myometrial tumors that are monoclonal and exhibit various forms of smooth muscle differentiation. The classic (conventional or typical) spindled form of leiomyoma is the most common solid pelvic tumor in females [1]. Occasionally, patients with a uterine mass presumed to be a classic leiomyoma are subsequently diagnosed with a benign leiomyoma variant (such as a cellular myoma) or a (malignant) uterine sarcoma, including endometrial stromal sarcoma, a malignant mixed mesodermal tumor, or a leiomyosarcoma. There is thus a broad spectrum of benign and malignant myometrial tumor variants. Some have a single histologic finding that is associated with leiomyosarcoma, such as an increased mitotic index or severe cytologic atypia or necrosis or pronounced hypercellularity, while other growth pattern variants are defined by their capacity to spread to peritoneal surfaces or solid organs but are histologically benign. Finally, some smooth muscle tumors do not fulfill the above-mentioned histologic criteria as being unequivocally benign or malignant and are, therefore, classified as smooth muscle tumors of uncertain malignant potential (STUMPs).

It is often a diagnostic challenge to distinguish a benign uterine fibroid from uterine sarcoma. Numerous studies have demonstrated the difficulty of establishing clear clinical criteria for differentiating uterine leiomyosarcomata from benign uterine leiomyomata. Imaging techniques such as ultrasound and magnetic resonance (MR) imaging can provide suggestions for the presence of uterine sarcoma, but they are not capable of definitively diagnosing it or ruling it out [2,3,4]. To date, more than 12,000 women with likely more than 30,000 uterine fibroids have been treated worldwide using transcervical fibroid ablation (TFA) with the Sonata^®^ System (Gynesonics, Redwood City, CA, USA). Sonata has been established as a highly effective, incisionless method for treating fibroids through the natural orifice of the cervix [5,6,7,8]. The Sonata System (Figure 1) integrates radiofrequency ablation electrodes and controls with a novel intrauterine ultrasound probe for real-time, high-resolution transcervical imaging from the device itself. Details and clinical experience with the Sonata System have been described [5,6,7,8].

The main teachings of this case include the fact that any nonresective fibroid treatment modality (embolization, medication, ablation) may be unknowingly performed on an undiagnosed uterine sarcoma. In addition, it is important to note that as TFA is not an intraperitoneal procedure and does not involve tissue morcellation or resection, there may not be a material risk of tumor dissemination should an unrecognized uterine sarcoma be ablated, thus preventing the spread of undiagnosed malignant cells, as might occur with laparoscopic power morcellation during myomectomy. This is the first published case report of TFA with the Sonata System in which the treated pathology was revealed to have been a leiomyosarcoma.

## 2. Case Presentation

A 42-year-old woman presented at a clinic in Switzerland with a chief complaint of heavy menstrual bleeding for more than one year. She had an established diagnosis of uterine fibroids, most notably an International Federation of Gynecology and Obstetrics (FIGO) type 2 submucous fibroid that was first diagnosed 4 years earlier. At the time of that original fibroid diagnosis, the myoma had measured 2.7 cm, but in the interim had increased in size to 3.8 cm (Figure 2). Perfusion on color Doppler was increased and was centrally located (Figure 3). The patient had been suffering from heavy menstrual bleeding for over a year and wished to preserve her fertility. Thus, a combination of transvaginal fibroid resectoscopy and transcervical fibroid ablation (TFA) with the Sonata System was chosen.

During hysteroscopy, the endometrial cavity was markedly indented by a type 2 fibroid. As the fibroid felt unusually soft and the Doppler sonography noted increased perfusion, a transcervical biopsy was taken with a resectoscope, after which two ablations were performed with the Sonata System. There were no other intracavitary lesions noted at hysteroscopy. The first ablation measured 3.3 cm × 2.4 cm, while the second ablation measured 2.6 cm × 1.9 cm. The patient’s immediate postoperative course was unremarkable. Histologic analysis of the sample taken was consistent with a STUMP. Immunohistochemistry revealed positive reactions for desmin, caldesmon, SMA, calponin, p16, fumarase hydratase, Ki67 (high, 40–50%), and p53 (wild type), with negative reactions for PanCK, CD10, ER, and S100.

Three weeks postoperatively, the patient experienced a vaginal expulsion of an approximately 2.5 cm diameter mass. The patient preserved the tissue, permitting histopathologic examination, confirming the hysteroscopic biopsy diagnosis of a STUMP. Magnetic resonance imaging (MR) of the pelvis revealed a 3.1 cm residual mass in the endometrial cavity without signs of myometrial invasion, along with a 0.7 cm calcified intramural fibroid. There was no sign of lymphadenopathy (Figure 4).

Transvaginal ultrasound displayed a 2.8 × 2.5 × 3.1 cm mass without signs of perfusion. Given the residual tumor mass that was ostensibly a STUMP but contained potential synchronous uterine sarcoma in the residual tissue, the patient opted to undergo laparoscopic hysterectomy six weeks after undergoing TFA with the Sonata System. Under the assumption that a uterine sarcoma might have been present, total laparoscopic hysterectomy with bilateral salpingectomy was performed, which proceeded uneventfully, corresponding to standard therapy of uterine sarcoma in Switzerland. Further measures (pelvic and/or paraaortic lymphadenectomy, infracolic omentectomy, radical hysterectomy, oöphorectomy) are not considered necessary according to the Deutsche Gesellschaft für Gynäkologie und Geburtshilfe (DGGG) guidelines. The patient was discharged on the second postoperative day. The patient will undergo regular clinical symptom-oriented gynecologic oncology follow-ups.

Histologically, the findings were consistent with a leiomyosarcoma (spindle cell “conventional” type). On microscopic examination, there was a significantly increased mitotic index, as well as tumor necrosis. The tumor demonstrated a spindle cell (conventional) morphology with compact and relatively well-organized fascicles with homogeneous cytological aspects without distinct cell atypia. Additional embedding showed a focal invasion pattern at an isolated site. The vital component was approximately 15%. The serosa was intact, and the resection borders were free of tumor cells. A polypoid intracavitary growth that was not seen during the original hysteroscopy showed no signs of malignancy.

## 3. Discussion

TFA with the Sonata System has been demonstrated in multiple studies to be a safe and effective method for treating nonpedunculated uterine fibroids (fibroids of FIGO types 1 through 6, including transmural fibroids). Transcervical radiofrequency ablation at 105 °C under integrated intrauterine ultrasound guidance in targeted fibroids leads to a volume reduction and a durable reduction in symptoms [8,9,10]. As with most alternatives to hysterectomy, this procedure is generally performed without a fibroid biopsy.

Numerous studies have demonstrated the difficulty of establishing clear and consistent clinical criteria for differentiating uterine leiomyosarcomata from benign uterine leiomyomata [2,3,4]. Some clinicians consider a possible diagnosis of uterine sarcoma when imaging suggests uterine fibroids but the patient manifests postmenopausal bleeding, abnormal premenopausal bleeding, and fibroid size of >5 cm in patients older than 45 years or despite GnRH-analog treatment [3]. Note that many of these symptoms and signs significantly overlap with (benign) fibroids. Indeed, approximately 5% of histopathologically benign fibroids can also exhibit sonographic features suggestive of malignancy [11]. There is a consensus, however, that no reliable diagnostic modality or algorithm can reliably rule out uterine sarcoma prior to hysterectomy. The term “rapid growth” has not been precisely defined in the medical literature and, in any event, has not been proven to represent an increased risk of uterine sarcoma [12].

Since our patient was under 45 years of age and presented with a presumptive myoma that had been present for 4 years, with only a slight increase in diameter from 2.7 cm to 3.8 cm, there was an understandable low suspicion for uterine sarcoma. This assumption is consistent with a prospective study by DeWaay and colleagues of 64 patients (mean age of 44 years) with benign fibroids, where the average growth rate was 1.2 cm in diameter over 2.5 years (range 0.9 to 6.8 cm) [13]. Therefore, a uterus-preserving method involving radiofrequency ablation was agreed upon by the patient and her physicians.

As per the 2020 World Health Organization (WHO) criteria, a STUMP consists of smooth muscle tissue with high cellular proliferation (20–22 mitotic figures/10 HPF), without significant atypia or necrosis, although it should be noted that there exist a range of histopathologic criteria for STUMPs, including the Stanford criteria. While STUMPs may recur in the absence of hysterectomy, because of the relative rarity of STUMPs, it is unclear how often they become truly malignant, although transformation to leiomyosarcomata has been reported [14,15]. Unfortunately, there are still no proven tumor markers that aid in the early diagnosis of uterine sarcoma, although genome profiling may be useful in distinguishing STUMPs from true uterine sarcomata [16].

Several studies have reported the prognostic factors for soft tissue sarcoma. Although serum lactate dehydrogenase (LDH) levels are associated with poor prognosis in several types of cancer, their role in soft tissue sarcomas remains unclear. Fujibuchi et al. evaluated the association between serum LDH levels and the clinical characteristics and prognosis of soft tissue sarcoma [17]. Therefore, in patients with high serum LDH levels at the first visit, these risks should be considered during pretreatment examinations and post-treatment follow-up. We did not perform an LDH test preoperatively on our patient because we had no suggestion for the presence of sarcoma.

In general, fibroid biopsy is not performed prior to transcervical fibroid ablation with the Sonata System but may certainly be performed at a physician’s discretion. In hindsight, the only feature consistent with uterine sarcoma was the increased internal vascularization on color Doppler ultrasound (Figure 3). Resistive index assessment was not performed. The value of diffusion-weighted MR imaging and other imaging techniques for distinguishing between uterine fibroids, STUMPs, and uterine sarcomata remains to be established, despite several promising studies [18,19,20].

At the time of the chosen procedure, hysteroscopic biopsy revealed a diagnosis of a STUMP. As the patient had a desire for future fertility and STUMPs most often do not exhibit malignant behavior, we discussed all possible scenarios with the patient [14,21,22,23]. It is important to note that STUMPs are rare, and there is no consensus regarding management, leaving limited, largely observational data as guidance [23].

In the end, the final diagnosis proved to have been stage IA leiomyosarcoma. The prognosis of leiomyosarcoma is very poor even in the early stages. Bacalbasa and colleagues reported only a 40% five-year survival for patients with FIGO stage I leiomyosarcoma [24].

We consider our patient to have been fortunate, as radiofrequency ablation with the Sonata System had been initially performed rather than laparoscopic myomectomy involving power morcellation. This is because fibroid morcellation (even with a scalpel) has been associated with intraperitoneal dissemination of both benign and malignant solid tumors and a worsened prognosis [25,26].

In summary, we have described a patient with stage IA leiomyosarcoma who was diagnosed only after presumptive fibroid expulsion status following transcervical fibroid ablation with the Sonata System. This is the first case report of a patient with uterine sarcoma diagnosed after TFA and is important for the following reasons:It serves as a reminder that any uterine mass may represent uterine sarcoma, including in premenopausal women.While very uncommon, STUMP lesions may progress to sarcoma or have a synchronous focus of malignancy.Because TFA does not involve tissue morcellation or resection, nor is it intraperitoneal, there may not be a material risk of tumor dissemination should an unrecognized uterine sarcoma be ablated.As with most patients treated with the Sonata System, postoperative surveillance may aid in a revised diagnosis of uterine sarcoma, hopefully in a timely fashion to not advance the surgical stage of the patient.

## 4. Conclusions

Uterine sarcoma remains a diagnostic dilemma and, while rare, is always a potential diagnosis in women of any age who present with putative uterine fibroids. Transcervical fibroid ablation, because it does not involve tumor resection or morcellation and is not an intraperitoneal treatment modality, does not appear likely to be associated with intraperitoneal seeding of benign or malignant tissue. While uterine sarcomata are fortunately rare entities, it is conceivable that appropriate surveillance after TFA may identify previously undiagnosed uterine sarcoma due to continued fibroid growth after ablation or, as in our patient, prolapse of the tumor through the external cervical os.

## Figures and Tables

**Figure 1 jcm-14-00088-f001:**
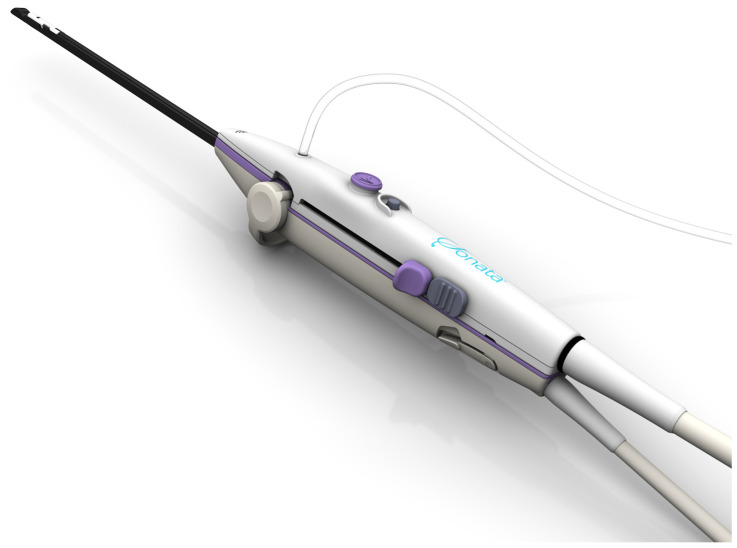
The Sonata handpiece. The top component contains the controls and delivery mechanism (Introducer and Needle Electrodes) for transcervical delivery of radiofrequency energy, while the bottom component consists of a novel intrauterine ultrasound probe for visualization and targeting. The system provides a graphical guidance system that includes an Ablation Zone and, uniquely among fibroid ablation devices, a Thermal Safety Border.

**Figure 2 jcm-14-00088-f002:**
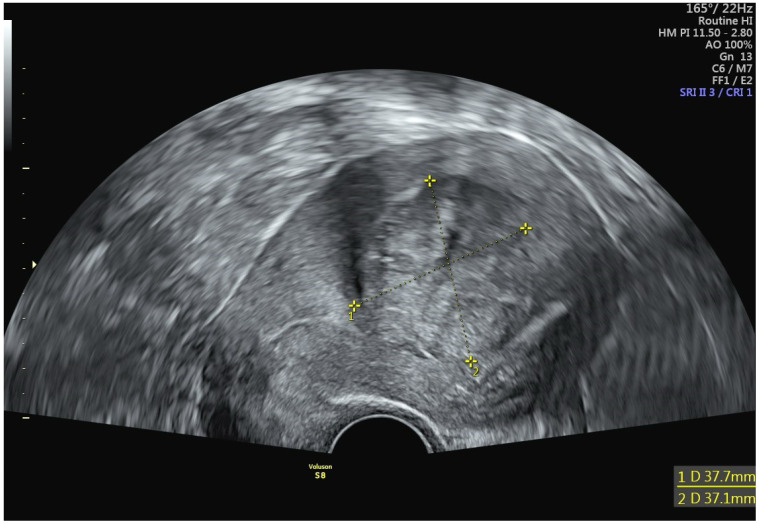
Transvaginal sonogram revealing a 3.8 cm FIGO type 2 myoma.

**Figure 3 jcm-14-00088-f003:**
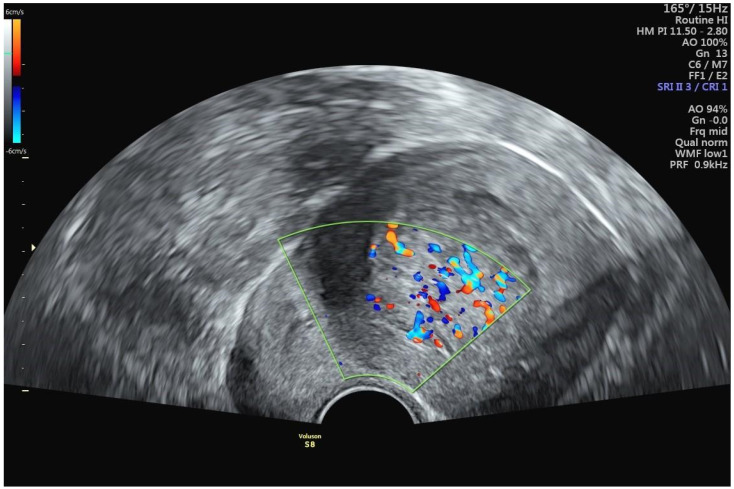
Color Doppler sonography of FIGO type 2 uterine fibroid, demonstrating unusual central (rather than peripheral) perfusion.

**Figure 4 jcm-14-00088-f004:**
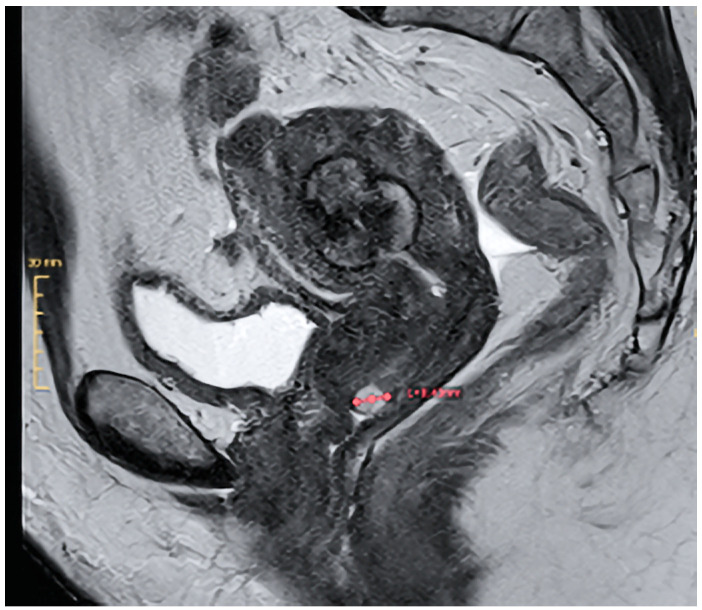
T2-weighted MR image demonstrating a 3.1 cm type 2 myoma and a small Nabothian cyst in the uterine cervix.

## Data Availability

The clinical data presented in this article are not readily available to maintain privacy of the treating physicians and their patient.

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
