# Peer review of "First Case Report of Uterine Leiomyosarcoma Diagnosed After Transcervical Fibroid Ablation"

_jcm, 2024, doi:10.3390/jcm14010088_

Round 1
Reviewer 1 Report
Comments and Suggestions for Authors
The SONATA ablation system may arise to mainstream fibroid treatment. The article in discussion bring to attention an important issue when treating fibroids. But the title is misleading as we learn from the case presentation that a biopsy was already performed during hysteroscopy that provided the STUMP result and that the mass expulsion was not bringing new results. Maybe it would have been useful if IHC would have been performed on both specimens. We also learn of supplemental pathology from the hysterectomy pathology exam - like the polypoid uterine cavity that should have been described at hysteroscopy. Also I think that the paper should better document pre-ablation ultrasound as the SONATA system is ultrasound guided and we are only informed of the ovarian endometrioma at the MRI before laparoscopy. I would like this work to show more ultrasound images before ablation and also procedural images. Again, ultrasound images preceding laparoscopic hysterectomy may be useful in completing the picture and enrich the readers' experience.
Introduction at line 43 has an unfinished sentence.
Author Response
Thank you very much for taking the time to review this manuscript. Please find the detailed responses below and the corresponding revisions/corrections highlighted/in track changes in the re-submitted files.
The SONATA ablation system may arise to mainstream fibroid treatment. The article in discussion bring to attention an important issue when treating fibroids. But the title is misleading as we learn from the case presentation that a biopsy was already performed during hysteroscopy that provided the STUMP result and that the mass expulsion was not bringing new results. Maybe it would have been useful if IHC would have been performed on both specimens. We also learn of supplemental pathology from the hysterectomy pathology exam - like the polypoid uterine cavity that should have been described at hysteroscopy. Also I think that the paper should better document pre-ablation ultrasound as the SONATA system is ultrasound guided and we are only informed of the ovarian endometrioma at the MRI before laparoscopy. I would like this work to show more ultrasound images before ablation and also procedural images. Again, ultrasound images preceding laparoscopic hysterectomy may be useful in completing the picture and enrich the readers' experience.
Thank you for your suggestions.
- The title has been changed to address your appropriate concerns
- The IHC findings have been added
- As the polypoid mass described in the hysterectomy specimen did not reflect an intracavitary polyp found during the original hysteroscopy (at the time of TFA), this has been clarified in the text (“There were no other intracavitary lesions noted at hysteroscopy.”)
- We have included an additional preoperative transvaginal scan (Figure 1) demonstrating the presumptive type 2 myoma that proved to have been a smooth muscle tumor of uncertain malignant potential.
- As the very small, possible endometrioma on MR was not identified in the preoperative transvaginal sonograms, and did not add anything to the topic at hand, this was removed from the text and the MR image is a slice that focuses of the residual mass in the endometrial cavity as this is the main focus of the case report
- We completely agree that preoperative imaging is helpful for teaching purposes and is one of the benefits of a case report!
Introduction at line 43 has an unfinished sentence.
Thank you. This was an inadvertent error during editing and has been removed.
Reviewer 2 Report
Comments and Suggestions for Authors
In this case report, the authors reported a case of TFA on a presumptive fibroid that was subsequently diagnosed as a STUMP tumor and ultimately a true leiomyosarcoma. Below are a number of issues that the authors shall address or revise:
1. No significance or novelty can be found in this case report. I am not sure what the authors want us to know in this case report.
2. No figure legend in figure 1 and 2.
3. The discussion part did not give us insights in TFA utility.
Comments on the Quality of English Language
This case report was not well organized and in bad writing, which was confusing.
1. Line 43, “It is known that MR imaging” was superfluous.
Author Response
Thank you very much for taking the time to review this manuscript. Please find the detailed responses below and the corresponding revisions/corrections highlighted/in track changes in the re-submitted files.
In this case report, the authors reported a case of TFA on a presumptive fibroid that was subsequently diagnosed as a STUMP tumor and ultimately a true leiomyosarcoma. Below are a number of issues that the authors shall address or revise:
- No significance or novelty can be found in this case report. I am not sure what the authors want us to know in this case report.
Thank you. This has been addressed in detail within the revised manuscript, including text within the Introduction and Discussion sections (in particular, the portion of the Discussion that states:
“This is the first case report of a patient with uterine sarcoma diagnosed after TFA, and is important for these reasons:
- It serves as a reminder that any uterine mass may represent uterine sarcoma, including in premenopausal women
- While very uncommon, STUMP lesions may progress to sarcoma or have a synchronous focus of malignancy
- Because TFA does not involve tissue morcellation or resection, nor is it intraperitoneal, there may not be a material risk of tumor dissemination should an unrecognized uterine sarcoma be ablated
As with most patients treated with the Sonata System, postoperative surveillance may aid in a revised diagnosis of uterine sarcoma, hopefully in a timely fashion to not advance the surgical stage of the patient.”
- No figure legend in figure 1 and 2.
This has been added, along with additional figures and legends
- The discussion part did not give us insights in TFA utility.
Additional information about TFA with the Sonata® System has been added to the Introduction, including an additional Figure.
Comments on the Quality of English Language
This case report was not well organized and in bad writing, which was confusing.
Thank you. This has been remedied in the text with several unnecessary sentences now removed, and many sentences simplified or corrected.
- Line 43, “It is known that MR imaging” was superfluous.
That partial sentence was an oversignt and has been removed. Thank you.
Round 2
Reviewer 1 Report
Comments and Suggestions for Authors
The authors took notice of the proposed changes and submitted the modified article so that I now consider the work proper for publication.
Reviewer 2 Report
Comments and Suggestions for Authors
I am satisfied with the author’s responses to my issues raised in my initial review. The revised manuscript is easier to follow based on feedback from the reviewers. I recommend that the revised paper be accepted.